# Inspiratory Muscle Training in Intermittent Sports Modalities: A Systematic Review

**DOI:** 10.3390/ijerph17124448

**Published:** 2020-06-21

**Authors:** Juan Lorca-Santiago, Sergio L. Jiménez, Helios Pareja-Galeano, Alberto Lorenzo

**Affiliations:** 1Faculty of Sport Sciences, Universidad Europea De Madrid, 28670 Madrid, Spain; jlorcasantiago13@gmail.com (J.L.-S.); helios.pareja@universidadeuropea.es (H.P.-G.); 2Sport Department, Facultad de Ciencias de la Actividad Física y del Deporte, Universidad Politécnica de Madrid, 28040 Madrid, Spain; alberto.lorenzo@upm.es

**Keywords:** inspiratory muscles, expiratory muscles, breathing exercises, sports performance, muscle strength

## Abstract

The fatigue of the respiratory muscles causes the so-called metabolic reflex or metaboreflex, resulting in vasoconstriction of the blood vessels in the peripheral muscles, which leads to a decrease in respiratory performance. Training the respiratory muscles is a possible solution to avoid this type of impairment in intermittent sports. The objective of this systematic review was to evaluate the results obtained with inspiratory muscle training (IMT) in intermittent sports modalities, intending to determine whether its implementation would be adequate and useful in intermittent sports. A search in the Web of Science (WOS) and Scopus databases was conducted, following the Preferred Reporting Elements for Systematic Reviews and Meta-Analyses (PRISMA) guidelines. The methodological quality of the articles was assessed using the PEDro (Physiotherapy Evidence Database) scale. In conclusion, the introduction of specific devices of IMT seems to be a suitable method to improve performance in intermittent sports, mainly due to a reduction of the metaboreflex, fatigue sensation, and dyspnea. The ideal protocol would consist of a combination of acute and chronic treatment, and, even if IMT is done daily, the duration will not exceed one hour per week.

## 1. Introduction

The definition of respiratory muscle training (RMT) itself is complex and confusing, mainly due to the terminology used. A distinction should be made between RMT and inspiratory muscle training (IMT), although some authors tend to use them analogously [1,2]. RMT includes both expiratory and inspiratory phases; the IMT includes only the inspiratory phase. In either case, both types of training seek to work a part or the whole of the respiratory muscles, which are the diaphragm, intercostals, abdominals, and in a minor way, the scalenus, sternocleidomastoid, trapezius, pectoralis major and minor, serratus, and supracostals [3,4]. IMT and RMT are useful in both healthy patients [5,6,7] and patients with pathologies [8,9].

In the past, it was thought that the respiratory system did not limit the capacity for aerobic physical work in healthy subjects [10]; however, it is now known that during high-intensity exercise, the respiratory muscles consume about ten to fifteen percent of total oxygen consumption [11], in addition to representing a cardiac output of fifteen percent of the total [12].

It can be assumed then, that fatigue in these muscles can cause great problems in the performance of athletes. This exhaustion causes a metabolic reflex (also called the metaboreflex), in which metabolites (i.e., hydrogen ions) accumulate in the respiratory muscles, a process that leads to an increase in sympathetic activity, causing, in turn, vasoconstriction in the peripheral locomotive muscles [13,14], reducing the blood flow of the extremities during exercise [15], leading to reduced exercise tolerance and increased dyspnea [16,17] and, therefore, a decrease in performance [14,18,19].

This metaboreflex originates as a consequence of the competition between the locomotive and respiratory muscles, both for blood flow and oxygen consumption during maximum exercise [10]. Respiratory muscle fatigue has been observed both in long-duration exercises at moderate intensity [20,21] and in those of short duration and high intensity [22].

This respiratory muscle fatigue was manifested by performing isocapnic hyperventilation at 70% of the maximum respiratory capacity, observing increases in blood lactate concentration, as seen in [10]. Similarly, it has been proven that using an assisted ventilator while exercising prevents diaphragmatic fatigue and increases blood flow to the lower limbs [23]; however, some authors argue that this fatigue does not produce decreases in performance [24].

In addition to biochemical and structural adaptations, training the inspiratory muscles would be useful to decrease the metaboreflex [25,26], by reducing fatigue in the respiratory muscles [27,28,29] through increased strength and endurance of the inspiratory muscles [30], in addition to lowering the lactate concentration [31,32,33,34]. All of this causes a reduction in both perceived effort [35,36] and the feeling of shortness of breath [37,38].

Two main methods of working the respiratory muscles have been distinguished, either by isocapnic hyperpnea [39,40,41] or by pressure threshold devices [2,42,43]. Concerning the first, isocapnic hyperpnea devices require air enriched in carbon dioxide, and the work is of low resistance (60–90% of maximum voluntary ventilation) at high speed, because the respiratory cycle is performed quickly. With the use of an isocapnic hyperpnea device, both inspiratory and expiratory muscles can be involved, which results in a greater capacity to perform intensive hyperpnea, that is, a higher resistance of the respiratory muscles [39,41,44]. On the other hand, pressure threshold devices do not require enriched air, and, besides, they run at a slower speed than the previous devices, and with higher resistance. In contrast to isocapnic hyperpnea, the expiratory musculature (except for the PowerLung device) does not work, although the inspiratory one does, causing as its main adaptation, an increase in the maximum inspiratory pressure (MIP). MIP is defined as the maximum pressure that can be carried out by the inspiratory muscles in a forced inspiration, which allows us to assess and compare the strength of the inspiratory muscles. These types of devices are easy to regulate, and can easily detect when the previously set pressure threshold loading is being exceeded [42,43,44]. It is important to note that, although it is thought that all pressure devices only work the inspiratory muscle (i.e., PowerBreathe), some also have a pressure threshold for the expiratory phase (i.e., PowerLung), thus working both phases, although some authors are against the latter due to the increase in intrathoracic pressure [45]. Resisted flow devices have been used to a lesser extent, because of the difficulty of controlling the inspiratory load, since there is no regulation mechanism [7].

Regarding the training protocol, a distinction must be made between the acute and chronic protocol. Acute training is a specific protocol that is carried out only before starting the tests. Its practical application would be in the warm-ups of both training and matches. In studies that have applied this type of protocol, two consecutive sessions of thirty breaths are performed in each session, with a one-minute recovery between sessions [46,47]. Chronic training consists of carrying out the work of the respiratory muscles throughout the day. Two sessions are usually performed daily, one in the morning and one in the afternoon, with thirty inspirations each [37,48]. Acute protocols are only applied on days when training and/or matches are held, and chronic protocols can be performed on any day of the week. Some studies used the two protocols in combination [49,50].

RMT has been shown to be useful in both short- and long-term continuous sports, such as athletics [51,52], swimming [41,43], cycling [39,42,53,54], and rowing [36]; however, there are far fewer studies related to intermittent sports modalities such as football [32,37,48] or rugby [38,47]. The intrinsic characteristics of this type of sports, such as the existence of a recovery time between efforts, which are frequently executed at a very high intensity and with a short duration, may be susceptible to improvement through RMT or IMT, both because of a possible decrease in the rating of perceived exertion (RPE) as well as in the sensation of dyspnea and, above all, by positively affecting the metabolic reflex. 

Other reviews related to this topic have been previously carried out [1,5], but they include either non-athletic subjects or non-intermittent sports modalities; therefore, the objective of the present study was to carry out a systematic review on the effect of RMT and IMT on intermittent sports modalities, assessing whether the results obtained justify its implementation, both acutely and chronically, in these sports modalities. 

## 2. Materials and Methods 

### 2.1. Study Design 

The present study was a systematic review, which used the methodological guidelines of the Preferred Reporting Items for Systematic Reviews and Meta-Analysis (PRISMA) approach [55].

### 2.2. Inclusion and Exclusion Criteria 

The PICOS question model was used to define the inclusion and exclusion criteria (Table 1). 

The inclusion criteria for selecting studies were as follows: (a) participants had to be athletes, both at amateur and professional levels; (b) they had to practice purely intermittent sports, as continuous efforts at varying intensities were not valid; (c) the study had to be a randomized controlled trial that compared a group performing either IMT or RMT with a placebo and/or control group; (d) only studies showing pre- and post-intervention results of all subjects were included; (e) articles were published in English; (f) only studies published in journals with an impact factor included in the Journal Citation Reports were admitted. Studies were excluded if: (a) the athletes had some type of physical and/or intellectual disability; (b) they were in the form of an editorial, letter to the editor, commentary, abstract, lecture, or opinion piece; (c) the assessment tests were not specific to the sport modality. A snowball sampling approach was made to find new studies.

In this systematic review, two authors (J.L.-S. and A.L.) independently searched the literature, reviewed titles and abstracts, and finally made a comparison between them, according to pre-determined inclusion and exclusion criteria. 

### 2.3. Search Strategy—Data Sources 

Searches were conducted on two databases, Web of Science (WOS) and Scopus, using the following search terms: (“Inspiratory training” OR “respiratory training” OR “inspiratory muscle training” OR “respiratory muscle training” OR “respiratory muscle endurance” OR “inspiratory muscle strength” OR “inspiratory muscle warm-up”) AND (“exercise” OR “intermittent exercise” OR “football” OR “soccer” OR “basketball” OR “badminton” OR “sport” OR “athletes” OR “sprint” OR “sprint-interval” OR “high-intensity exercise”), including articles up to and including 20 March 2020. All items were referenced for manual duplicate removal. 

Once the duplicates were removed, the title and summary of the remaining studies were revised. Subsequently, thirty-three studies were evaluated in full text, excluding those that did not meet the inclusion or exclusion criteria. Finally, a qualitative and quantitative analysis of the studies admitted to the systematic review was carried out. 

### 2.4. Methodological Quality of the Studies 

To assess the methodological limitations of the papers and to be able to compare them, an assessment of the methodological quality of all studies was carried out using the PEDro (Physiotherapy Evidence Database) scale (Table 2), which is proven to be reliable for assessing randomized controlled trials [56]. This scale consists of 11 items related to eligibility criteria, random allocation, concealed allocation, follow-up, baseline comparability, blinded subjects, blinded therapists, blinded assessors, intention to treat, between-group analysis, and both point and variability measures. The maximum score is 10 points, because item 1, which is an eligibility criterion, only affects the external validity [57].

Methodological quality testing was independently performed by two reviewers (J.L.-S. and A.L.) and, in case of disagreement, a third reviewer (S.L.J.) was consulted to make a decision. According to the PEDro scale, the studies obtained a methodological quality of between 50–90%, as such, no article was excluded for failing to achieve a minimum rate, which in this case would be to obtain a score of less than 4 points (40%). The study was considered to be excellent quality if the PEDro score was 9 or 10, good quality if the score was from 6–8, fair quality if the score was 4 or 5, and finally, poor quality if the score was 3 or lower. 

### 2.5. Data Collection and Extraction

Once the final studies were obtained, the essential information was reported in three different tables, which included the following items: main author and year of publication, characteristics of the participants (number of subjects and their gender, age, and maximum oxygen consumption), sports context (the type of sport practiced and its sports level), description of the interventions (the kind of training, the initial intensity with its respective progression, sessions per week, number of weeks, duration of exercise, assessment of whether it was supervised at all times, and a brief description of how to proceed with the control group and/or placebo), and description of the results (the specific test used, variables analyzed, and main results obtained). 

## 3. Results

### 3.1. Study Selection

The search provided a total of 344 articles, 180 from WOS and 164 from Scopus. One hundred seventy-two duplicates were eliminated after a manual analysis was conducted. Of the remaining 172, a total of 139 were excluded: (a) 99 articles were deleted because the title was not consistent with the subject matter; (b) 31 articles were related to continuous sports modalities; (c) 8 articles were reviews or a meta-analysis; (d) 1 article was on non-athletes. Previous reviews and meta-analyses were reviewed to detect possible studies eligible for the current systematic review. 

The full text of 33 articles was evaluated and 23 were excluded for the following reasons: (a) 8 articles, despite including intermittent modalities, used a non-specific evaluation test; (b) 7 articles included people with some type of disability; (c) 5 articles did not have a placebo or control group; (d) 1 article was not in English; (e) 1 article did not specify whether there was randomization in the groups; (f) 1 article only had a conference abstract.

Thus, a total of 10 articles were finally included in the systematic review (Figure 1). The reference lists of these 10 studies were reviewed to find more articles that could be included in this work.

### 3.2. Characteristics of the Participants

A total sample of 218 was obtained from all the articles, divided into 193 men and 25 women, all of whom were young. The most studied sports were soccer and rugby, although badminton, field hockey, and basketball were also studied to a lesser extent. Regarding the sports level of the subjects, two studies included professional athletes, two articles included semi-professional athletes, and seven studies were made up of recreational players. 

Table 3 provides more detailed information about the characteristics of the sample. 

### 3.3. Description of the Interventions and the Results

In all interventions, a training device based on pressure thresholds was used without using isocapnic hyperpnea or resisted flow. The initial effort intensity was very similar in all studies, between 40–60% of the MIP, except for one article that used 80% of the MIP from the beginning.

Three types of intervention were used, namely: (a) IMT in a chronic protocol (*n* = 7) [32,37,38,48,58,59,60]; (b) in an acute protocol before the evaluation test (*n* = 2) [46,47]; (c) with a combination of chronic and acute protocols (*n* = 1) [50]. The chronic intervention procedure lasted between five and twelve weeks so that no homogeneity can be seen in them. Acute training studies made up 40% of the MIP, thirty inspirations, and two sessions before the test. 

The number of weekly sessions was heterogeneous. Although two sessions of the inspiratory muscle warm-up were always performed before the test, the chronic treatment differed significantly from one article to another, ranging from three sessions to fourteen per week. 

The level 1 Yo-Yo Intermittent Recovery test was used as an evaluation test to compare pre- and post- IMT effects in 6 out of 10 articles. One of the studies that used the Yo-Yo test also assessed the effects of the performance obtained in high-intensity intervals. Of the other four articles, two used a repeated sprint ability (RSA) test, one used a football-specific fitness test protocol (FSFT), and one used a badminton-footwork test. 

A significant increase in MIP was obtained in all studies, ranging from an improvement of 8% to 33%. The RPE and rate of perceived dyspnea (RPB) significantly decreased in the studies investigated. Nine of the ten articles showed a significant improvement in performances for their respective assessment tests, referring to a reduction in sprinting time, a greater recovery capacity, and a greater distance covered or speed in sprinting. 

Table 4 shows, in detail, the interventions and results achieved in each study.

## 4. Discussion

The objective of the present study was to evaluate the effects of RMT and IMT on intermittent sports modalities. After the detailed examination of the ten articles that make up this systematic review, with a sample of 218 subjects, it seems clear that the use of IMT is useful in producing performance improvements in intermittent sports modalities, as reflected in the results of the Yo-Yo test, the RSA, and the specific badminton test. It should be noted that performance increases ranged from 4% [48] to 53% [50]. Only one study [32] did not observe an increase in performance, although they noticed a rise in MIP and a decrease in blood lactate; however, in this study, IMT was performed only two days a week, which may not be sufficient to obtain adaptations with this type of training [32].

Concerning the methodological quality of the articles analyzed, the average rating of all papers is 6.7 points, which is a “good” score. This score is similar to that obtained in other systematic reviews of the same topic [1,5]. The main methodological problems encountered by our review are related to the non-concealed allocation to groups and blinding of subjects, researchers, and evaluators. It would be interesting if these factors were taken into account in future studies to achieve greater methodological quality.

### 4.1. Protocol and Intervention

#### 4.1.1. Chronic Treatment

Most studies conducted a chronic treatment of IMT, ranging from 5–12 weeks. The heterogeneity in the number of weekly sessions is very relevant since there are protocols that range from three weekly sessions [60] to fourteen [58]. Despite the considerable differences between them, the vast majority showed significant increases in performance. It is also not clear that doing more sessions per week brings more benefits, since in the study by Nicks et al. with a sample of professional players [59], despite having fewer weekly sessions and fewer total number of weeks, a higher performance was obtained than in the study by Romer et al. [58] with recreational and semi-professional players (≈17% vs. ≈7%). A possible explanation for this difference is that the professional players used the PowerLung device, which allows for the training of both expiratory and inspiratory muscles, whereas the recreational and semi-professional group used the POWERbreathe device, which works only on the inspiratory phase. Future research should study the different results between one device and another.

#### 4.1.2. Acute Treatment

Two studies conducted an acute treatment [46,47], carrying out the same protocol in a strictly accurate manner, with a 40% resistance of the MIP, performed in two sessions before the test, adding up to a total of 60 inspirations, and comparing it with both a control and a placebo group, which followed the same protocol, but at 15% of the MIP, a value that has previously been shown to not generate any effect [47]. Using 40% of the MIP (≈7% vs. ≈19%), both studies observed a significant increase in performance in the experimental group. Other variables such as MIP and dyspnea also improved significantly in these studies.

Other studies have not observed an improvement produced by the warming of the inspiratory muscles [61,62]; however, these studies were conducted in continuous sports modalities, which could explain the difference in results. Further studies on the possible beneficial effects of acute IMT in intermittent sports would be needed.

#### 4.1.3. Combination of Acute and Chronic Treatment

Only the study by Tong et al. [50] conducted a combined acute and chronic protocol. It consisted of running four weeks of IMT before intervallic sprint testing, which was preceded by a warm-up of the inspiratory muscles. Although the players, coming from both football and rugby, were not professionals, they trained very often, averaging two or three hours a day, four or five days a week. With this in mind, it is surprising to see the performance improvements in both the Yo-Yo test (≈31%), and the percentage variation in the ability to maintain high speeds over different distances (200 m, ≈25%; 600 m, ≈53%; 800 m, ≈36%). Another study, not included in the systematic review because it did not specify whether the division of the groups was randomized or not, supports the idea that the combination of chronic and acute training provides the greatest benefits, demonstrating that the group performing both protocols obtained the best results, followed by the performance of chronic IMT and, finally, the performance of acute IMT [49].

### 4.2. Sports Practiced and Sports Level

Most of the studies included in the systematic review focused primarily on two sports, football and rugby. Athletes from other sports, such as basketball, badminton, and field hockey also appeared in some studies. Most of them included recreational subjects, although two studies had semi-professionals and another two included professional athletes.

Even though the chronic protocols differed significantly in the number of weekly sessions and total weeks of the proposed protocol, the values of the intensity for the MIP were around 40% and 50%; therefore, it is worth noting that the article by Nunes Júnior et al. [60] was the only one included in this review that used a much higher intensity, precisely 80% of the MIP, obtaining very positive values, such as an increase in performance in the Yo-Yo test of ≈14% and a maximum ventilatory volume of 22%. Future research could be directed at assessing IMT with a very high load in higher-level athletes, as it has been studied very little [52,63,64].

All subjects experienced an increase in maximum expiratory pressure (MEP), ranging from a 9% to 32% improvement, and, it is worth noting that the increase in MEP in the article by Nunes Júnior et al. [60] is quite surprising because the device they used was solely and exclusively for the work of the inspiratory muscles. This even exceeded the increase in MEP (≈32% vs. 29%). It is important to note that many of the other articles did not assess MEP, but for those that did, they did not observe any significant increases. It would be interesting to determine whether the work of the inspiratory muscles can favor the work of the expiratory ones, as this seems to be indicated in an article with semi-professional handball players where the MIP increased by 54% and the MEP by 23% [65]. Only one study used PowerLung, a device that presents a load on both the inspiratory and expiratory phases [59]; however, since it did not provide MEP data, a comparison could not be made.

Those who evaluated the decrease in lactate concentration had mixed results. Although professional subjects did not observe any reduction, those who were not professionals did have a significant decline of approximately 30%. A possible explanation for this would be that those athletes with a non-professional level could benefit more from a decrease of the metaboreflex, which would not be so significant in subjects with an expert level, as they had more developed respiratory muscles indirectly, due to more hours of physical training.

The vast majority of studies evaluating the rating of perceived dyspnea (RPB) were very positive, except for the article by Guy et al. [32]; however, there is a potential explanation for the non-appearance of positive results, since this study, as the authors themselves acknowledge, used a deficient number of days per week of IMT, with only two days per week. This protocol clashes with the vast majority of other studies, which proposed five to seven days per week. Furthermore, it was also not reported whether the studies were supervised, and there was no progressive increase over the six-week protocol. It was also the only study with a chronic IMT that also did not observe improvements in performance, nor reductions in the RPE.

Although the majority of chronic studies that analyzed RPE observed very significant decreases, from 8% [50] to 29% [37], with a large effect size (ES) (−1.47), acute training protocols did not observe a reduction. This last value is very remarkable, as the subjects were semi-professional players.

The findings from semi-professional or professional participants have to be separately interpreted. Among the first, in the article by Najafi et al. [37], increases of 8.9% were achieved with a moderate ES of 0.77 in experimental group 1 (55% MIP), and 8.1% with a moderate ES of 0.78 in experimental group 2 (40% MIP); however, in the article by Romer et al. [58], the increase was approximately 7% with a moderate ES of 0.25. Both studies are in line with other studies conducted with semi-professional handball players [65] and football players [2], which showed performance improvements.

Concerning the analysis of the studies that implemented an IMT protocol in professional athletes, it should be noted that all the subjects were footballers [48,59]. Archiza et al. [48] evaluated the possible improvements of IMT through RSA and observed that the best sprint time was reduced by approximately 4% with moderate ES, the average time between all sprints by ≈6%, and a reduction of the performance decrease between sprints by around 30% with large ES, in addition to increasing the MIP by approximately 22%, all of which are impressive values. In the study by Nicks et al. [59] an increase of ≈20% of the MIP and an improvement of the performance of about 17% in a test of RSA was observed. These studies reinforce the results obtained in an article with first-division players of the Brazilian football league [30] and another with professional handball and basketball players [63].

There are three main factors for this improvement in both the Yo-Yo and the RSA tests in semi-professional and professional athletes: first, a reduction in metaboreflex, which would allow these athletes to get more oxygen to the peripheral muscles; second, a decrease in the RPE; third, a reduction in the RPB.

It is difficult to compare the results obtained in semi-professional and professional players to recreational players, mainly due to the great difference in the protocols used. Despite this, logically speaking, it would be reasonable to expect that there would be a more remarkable improvement in recreational athletes, since they would have worked their respiratory muscles less, whereas semi-professional and professional players would have developed them more via physical exercise and training.

The particularity of intermittent sports, fundamentally characterized by the recovery time between efforts, could be one of the causes that justify these meaningful performance increases. If the athlete perceives a lower rate of work and a lesser feeling of shortness of breath, it will enable him/her either to recover after from the previous effort, and to carry out another one, or to increase performance in the subsequent actions. Two studies that could demonstrate this reasoning are Archiza et al. [48], who evaluated the performance between the different sprints, improving by about 30%, and Tong et al. [50], who also found that athletes who underwent inspiratory training were able to perform each sprint at a higher speed than the control group. Some studies show that a reduction in dyspnea is directly related to improved performance in badminton players [66]. In short, it seems clear that if an athlete perceives a lesser feeling of dyspnea, he will be able to carry out efforts with a smaller margin of recovery or, conversely, carry out continuous efforts at a greater speed. Either case provides a competitive advantage.

Similarly, as it is not a typical training protocol, its implementation can lead to a greater increase in performance, mainly because of the motivating effect of novelty.

The results of this systematic review are in line with the systematic review of HajGhanbari et al. [5], although their work is focused more on continuous sports modalities; and also in line with that of Sales et al. [1], although in this study they focused more on assessing the resistance of respiratory muscles, without going into a detailed evaluation of sports performance improvement. These two systematic reviews used different training devices, both pressure threshold and isocapnic hyperpnea, and in the study by Sales et al. [1], non-sporting subjects were included. In the present review, only previous investigations using the pressure threshold devices with athletes were considered.

### 4.3. Study Limitations

This systematic review has several limitations. Firstly, the low number of papers related to this topic makes it difficult to compare the results obtained in some sports, such as basketball or hockey.

Another major problem was the assessment of the sporting level of the athletes, due to the different ways in which the studies expressed this aspect. The fact of including only articles in English in the systematic review means that some interesting papers in other languages were excluded.

The decision to include only those studies with specific evidence of assessment of the sports modality may have excluded some interesting articles, although this has given more relevance to this systematic review, as well as only selecting randomized clinical trials. The choice of excluding non-specific evaluation tests is justified by the fact that, in this systematic review, we sought direct practical applications, considering that it is not useful for players who play football or basketball to perform tests on a cycle ergometer [67], to perform tests such as Cooper’s [2], or tests without recovery between efforts [68], as they have no direct applicability in their respective sports.

The last, and probably the biggest limitation, was the low sample size that we obtained (218 subjects) not only from the studies as a whole, but also individually. This could be a consequence of the fact that, although RMT and IMT were adopted many years ago, their application has been more focused on sports modalities characterized by continuous effort, since it was thought that respiratory muscle fatigue was only possible in this type of sport; however, this point has already been refuted, and it has been shown that respiratory muscle fatigue exists in intermittent modalities.

Because of these limitations, the results obtained in this systematic review should be treated with caution, although some results are encouraging, such as the increase in MIP or the reduction in the rate of perceived effort and dyspnea.

## 5. Conclusions

The introduction of specific devices to work the respiratory muscles, and in particular, the inspiratory muscles seems to be useful in intermittent sports modalities to improve performance, mainly due to a reduction of metaboreflex, fatigue sensation, and dyspnea.

The ideal protocol would consist of a combination of acute treatment, performing two consecutive sessions of 30 inspirations at 40% of the MIP with a minute rest between the two sessions, and chronic treatment, with two sessions per day, four to seven times a week, at an intensity of 50% of the MIP.

Although three main types of devices perform respiratory and IMT, the advantages of pressure threshold devices make them ideal for use.

The use of this type of training is recommended in any intermittent sports modality due to the high dose–response relationship.

Future studies should be directed at observing possible performance improvement in professional athletes, as there are few studies available. It would also be interesting to study how the use of higher intensities, close to 80–85% of MIP, affects inspiratory training.

## 6. Practical Applications

Although the inclusion of this type of training can be complicated, especially in an acute form as there is not enough time for warm-up, its application in semi-professional and professional athletes would be highly recommended.

One possible application of the acute protocol could be in the moments before the start of the warm-up, when the players are in the locker room, as its use does not require great concentration or the use of hands. This warm-up would take approximately ten minutes, performing two sessions of 30 breaths each, with a minute rest between them, at 40% of the MIP.

Regarding the chronic protocol, the ideal procedure would be to carry out two sessions of 30 inspirations at 50% of the MIP, one session in the morning and one in the afternoon, between four and seven times a week. Athletes could be asked to use the device while doing everyday activities, as it does not require much concentration, two sessions could be completed in just ten minutes.

A practical example is provided in Table 5, as to how IMT could be utilized in a game of professional football.

There would be five days of chronic IMT and two days of acute IMT. Even if they are performed every day of the week, the weekly duration will, in no case, exceed one hour of work, assuming that 60 inspirations per day are carried out in less than ten minutes.

Another possible application of this type of training could be when an athlete is in a period of injury. This will ensure the maintenance of training adaptations of the respiratory muscles, which will lead to a better physical condition when returning to training.

Similarly, the existence of responders and non-responders to this type of work should be assessed, in addition to taking into account that, once the appropriate adaptations have been obtained, the frequency of this type of training can be reduced to a third of what was being done, since it would not worsen the inspiratory function [69].

## Figures and Tables

**Figure 1 ijerph-17-04448-f001:**
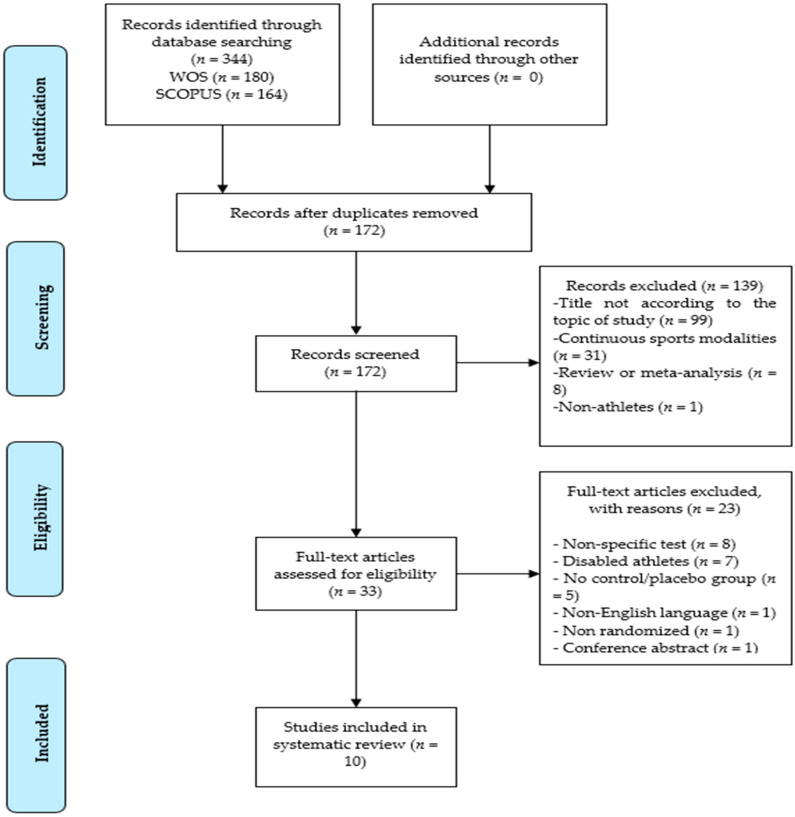
Preferred Reporting Elements for Systematic Reviews and Meta-Analyses (PRISMA) flow diagram.

**Table 1 ijerph-17-04448-t001:** PICOS question model.

P (Population)	I (Intervention)	C (Comparison)	O (Outcomes)	S (Study Design)
Subjects who practice intermittent sports modalities	Use of RMT and/or IMT	Same conditions with placebo and/or control groups	Increase in performance	Randomized controlled trial (RCT)

RMT: respiratory muscle training; IMT: inspiratory muscle training.

**Table 2 ijerph-17-04448-t002:** PEDro (Physiotherapy Evidence Database) scale quality assessment *.

First Author, Year	1	2	3	4	5	6	7	8	9	10	11	Total	Quality
Romer, 2002 [58]	Yes	1	1	1	1	1		1	1	1	1	9	Excellent
Tong, 2006 [47]	Yes	1		-	-			1	1	1	1	5	Fair
Lin, 2007 [46]	Yes	1	-	-	-			1	1	1	1	5	Fair
Tong, 2008 [38]	Yes	1		1	1			1	1	1	1	7	Good
Nicks, 2009 [59]	Yes	1		1				1	1	1	1	6	Good
Tong, 2010 [50]	Yes	1		1				1	1	1	1	6	Good
Guy, 2014 [32]	Yes	1	-	1	1	-		1	1	1	1	7	Good
Archiza, 2017 [48]	Yes	1		1	1	1		1	1	1	1	8	Good
Nunes Júnior, 2018 [60]	Yes	1	1	1	-			1	1	1	1	8	Good
Najafi, 2019 [37]	Yes	1		1				1	1	1	1	6	Good

* 1 = eligibility criteria were specified, 2 = subjects were randomly allocated to groups, 3 = allocation was concealed, 4 = the groups were similar at baseline regarding the most important prognostic indicators, 5 = there was blinding of all subjects, 6 = there was blinding of all therapists who administered the therapy, 7 = there was blinding of all assessors who measured at least one key outcome, 8 = measures of at least one key outcome were obtained from more than 85% of the subjects initially allocated to groups, 9 = all subjects for whom outcome measures were available received the treatment or control condition as allocated or, where this was not the case, data for at least one key outcome was analyzed by “intention to treat”, 10 = the results of between-group statistical comparisons are reported for at least one key outcome, 11 = the study provides both point measures and measures of variability for at least one key outcome.

**Table 3 ijerph-17-04448-t003:** Characteristics of participants *.

First Author, Year	M/F	Age in Years (SD)	VO_2max_ (mL/kg/min)	Sport	Competitive Level
Romer, 2002 [58]	24/0	E = 21.3 (1.1) P = 20.2 (0.7)	E = 56.3 (0.9) P = 55.8 (1.7)	Football, rugby, field hockey, and basketball	Recreational and semi-professional
Tong, 2006 [47]	10/0	T = 21.3 (1.2)	T = 62.9 (4.2)	Football and rugby	Recreational
Lin, 2007 [46]	10/0	T = 23.0 (2.0)	T = 51.0 (6.0)	Badminton	Recreational
Tong, 2008 [38]	30/0	E = 21.3 (0.9) P = 21.5 (2.1) C = 22.0 (1.9)	E = 60.8 (4.7) P = 55.8 (7.9) C = 59.1 (5.2)	Football and rugby	Recreational
Nicks, 2009 [59]	20/7	E = 19.8 (0.9) C = 19.9 (1.3)	Not reported	Football	Professional
Tong, 2010 [50]	18/0	E = 21.1 (1.1) C = 22.3 (1.0)	E = 59.0 (6.3) C = 58.1 (4.5)	Football and rugby	Recreational
Guy, 2014 [32]	31/0	E = 26.6 (8.2) P = 23.9 (6.7) C = 21.3 (4.9)	E = 44.0 (6.7) P = 42.9 (8.7) C = 46.3 (6.2)	Football	Recreational
Archiza, 2017 [48]	0/18	E = 22.0 (3.9) P = 20.1 (2.0)	E = 41.2 (4.0) P = 41.7 (3.8)	Football	Professional
Nunes Júnior, 2018 [60]	20/0	E = 22.0 (4.0) C = 23.0 (2.0)	Not reported	Rugby	Recreational
Najafi, 2019 [37]	30/0	E1 = 16.5 (0.7) E2 = 16.7 (0.5) P = 16.7 (0.8)	Not reported	Football	Semi-professional

* The data are reported as mean (SD). M/F = male/female; E = inspiratory muscle training group; P = placebo group; C = control group; T = total sample.

**Table 4 ijerph-17-04448-t004:** Description of the interventions and the results.

First Author, Year	Type of Training	Starting Intensity	Progression of Training Intensity	Number of Sessions per Week	Number of Weeks	Duration of Exercise	Supervision	Control/Placebo	Specific Test	Main Results of the Analyzed Variables
Romer, 2002 [58]	POWERbreathe (threshold)	50% MIP	Progressive increase, until they can only do 30 repetitions	2 sessions daily, 7 days per week	6 weeks	30 inspirations	Supervised	Placebo: 1 session of 60 inspirations at 15% MIP	RSA	↑ PIF (≈20%) ↑ MIP (≈33%) ↑ Performance (≈7%)
Tong, 2006 [47]	POWERbreathe (threshold)	40% MIP	No progression	2 pre-test sessions	Only 1 test	30 inspirations	Supervised	Placebo: same protocol, but at 15% MIP Control: no intervention	Yo-Yo intermittent recovery test	↑ MIP (≈9%) ↑ Vmax (≈5%) ↑ WImax (≈21%) ↑ Popt (≈16%) ↑ MRPD (≈13%) ↓ RPB/4i (≈22%) ↑ Performance (≈19%)
Lin, 2007 [46]	POWERbreathe (threshold)	40% MIP	No progression	2 pre-test sessions	Only 1 test	30 inspirations	Supervised	Placebo: same protocol, but at 15% MIP Control: no intervention	Badminton-footwork test	↑ P_0_ (≈8%) ↑ MRPD (≈9%) ↓ RPB/min (≈7%) ↑ Performance (≈7%)
Tong, 2008 [38]	POWERbreathe (threshold)	50% MIP	Increase of 10 or 15 cmH_2_O when 30 repetitions are performed without stopping	2 sessions daily, 6 days per week	6 weeks	30 inspirations	Supervised	Placebo: same protocol, but at 15% MIP Control: no intervention	Yo-Yo intermittent recovery test	↑ P0 (≈32%) ↑ WImax (≈40%) ↑ Popt (≈38%) ↑ MRPD (≈39%) ↓ RPE/4i (≈11%) ↓ RPB/4i (≈12%) ↓ 20 m VE (≈10%) ↓ 20 m VT/ti (≈8%) ↑ Performance (≈16%)
Nicks, 2009 [59]	PowerLung (threshold)	50% MIP	Progressive increase once or twice a week, until they can only do 30 repetitions	2 sessions daily, 5 days per week	5 weeks	30 inspirations	Normally supervised, when not, participants submitted training logs	Control: no intervention	RSA	↑ MIP (≈20%) ↑ Performance (≈17%)
Tong, 2010 [50]	POWERbreathe (threshold)	E_IMT_ = 50% MIP E_WU_ = 40% MIP	Increase of 10 or 15 cm H_2_O when 30 repetitions are performed without stopping	E_IMT_ = 2 sessions daily, 6 days per week E_WU_ = 2 pre-test sessions	E_IMT_ = 4 weeks E_WU_ = 6 weeks	30 inspirations	Not reported	Control: no intervention	Yo-Yo intermittent recovery test	↑ P0 (≈20%) ↓ RPE (≈8%) ↓ RPB (≈16%) ↓ 20 m VE (≈10%) ↓ 20 m VE/VO2 (≈8%) ↓ 20 m VT/Ti (≈10%) ↓ 10 s VE (≈10%) ↓ 10 s VE/VO2 (≈5%) ↑ Performance1 (≈31%) ↑ Performance2 (200 m, ≈ 25%; 600 m, ≈ 53%; 800 m, ≈ 36%)
Guy, 2014 [32]	POWERbreathe (threshold)	55% MIP	No progression	2 sessions daily, 2 days per week	6 weeks	30 inspirations	Not reported	Placebo: same protocol, but at 15% MIP Control: no intervention	SSFT	↑ MIP (≈13%) ↓ Blood lactate (≈32%)
Archiza, 2017 [48]	POWERbreathe (threshold)	50% MIP	Progressive increase every week	2 sessions daily, 5 days per week	6 weeks	30 inspirations	Supervised	Placebo: same protocol, but at 15% MIP	RSA	↑ MIP (≈22%) ↓ RSA_BEST_ (≈4%) ↓ RSA_MEAN_ (≈6%) ↓ RSA_DEC_ (≈30%)
Nunes Júnior, 2018 [60]	Breather Plus IMT Power (threshold)	80% MIP	Progressive increase from the fourth training session	3 sessions per week	12 weeks	30 inspirations	Supervised	Control: same protocol but without resistance	Yo-Yo intermittent recovery test	↑ MVV (22%) ↑ MIP (≈29%) ↑ MEP (≈32%) ↑ Performance (≈14%)
Najafi, 2019 [37]	POWERbreathe (threshold)	E1 = 55% MIP E2 = 45% MIP	Progressive increase once a week	2 sessions daily, 5 days per week	8 weeks	E1 = 25–35 inspirations E2 = 45–55 inspirations	Supervised	Placebo: 30 inspirations at 15% MIP	Yo-Yo intermittent recovery test	↑ MIP (E1: 27.2%; E2: 30.6%) ↓ RPE (E1: 26.9%; E2: 28.9%) ↓ RPB (E1: 62.1%; E2: 56.3%) ↓ Lactate (E1: 29.4%; E2: 27.5) ↓ Fatigue index (E1: 34.4%; E2: 40.6%) ↑ Performance (E1: 8.9%; E2: 8.1%)

* MIP = maximal inspiratory pressure, RSA = repeated sprint ability, PIF = peak inspiratory flow, Vmax = maximal inspiratory flow, WImax = maximal inspiratory muscle power, Popt = optimal pressure, MRPD = maximum rate of pressure development, RPB = rating of perceived dyspnea, P0 = maximal inspiratory pressure at zero flow, RPE = rating of perceived exertion, 20 m = sprint of 20 m, VE = minute ventilation, VT = tidal volume, Ti = inspiratory time, EIMT = chronic inspiratory muscle training, EWU = acute inspiratory muscle training (warm-up), RSABEST = RSA best performance time, RSAMEAN = RSA mean performance time, RSADEC = RSA performance decrement, MVV = maximum voluntary ventilation, MEP = maximal expiratory pressure, SSFT = soccer-specific fitness test protocol, E1 = IMT group number one, E2 = IMT group number two, ↑ = significant increase over control group (*p* < 0.05), ↓ = significant decrease over control group.

**Table 5 ijerph-17-04448-t005:** Example microcycle using IMT.

Monday	Tuesday	Wednesday	Thursday	Friday	Saturday	Sunday
**Chronic IMT**, performing one session in the morning and another at night	**Chronic IMT**, performing one session in the morning and another at night	**Chronic IMT**, performing one session in the morning and another at night	**Acute IMT**, performing both sessions in the locker room, before going out to train	**Chronic IMT**, performing one session in the morning and another at night	**Chronic IMT**, performing one session in the morning and another at night	**Acute IMT**, performing both sessions in the locker room, before going out to pre-game warm-up
Training	Rest	Training	Training	Training	Activation	Match

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
