# Peer review of "Inspiratory Muscle Training in Intermittent Sports Modalities: A Systematic Review"

_ijerph, 2020, doi:10.3390/ijerph17124448_

Round 1
Reviewer 1 Report
I have read the article by Lorca et al. with great interest. The article is well written. I have only some minor comments:
- Line 36. I guess the sentence applies in healthy people. Patients with respiratory disease have always been thought to have some limitation in their exercise capacity.
- Line 42. Please give some specific examples what metabolites you are talking about.
- It should be clearly stated in the Introduction if any systematic reviews had been conducted in this topic and if so, what this article adds to the current knowledge.
- Study selection. I wonder if the authors looked at the references list of the selected studies and checked articles which referred to these studies.
Author Response
Dear Reviewer,
Thank you very much for your comments and for the details that you have commented and we have modified.
In addition, at the request of other reviewers we have conducted a review of English. We attach certificate.
In relation to this, we have modified many of the pages of the previous article, so you will find your comment in the main text as reviewer 1.
Line 36. I guess the sentence applies in healthy people. Patients with respiratory disease have always been thought to have some limitation in their exercise capacity.
- You are absolutely right, we have reflected it in the study
Line 42. Please give some specific examples what metabolites you are talking about.
- We have taken the example of the hydrogen ions.
It should be clearly stated in the Introduction if any systematic reviews had been conducted in this topic and if so, what this article adds to the current knowledge.
- We have reflected two systematic reviews; although we had reflected it in the discussion section, we forgot to put it in the introduction. Thank you.
Study selection. I wonder if the authors looked at the references list of the selected studies and checked articles which referred to these studies.
- We have nuanced the reading of reviews and meta-analyses, and the articles were chosen to see their list of references. Despite this, we did not find results that met the inclusion and exclusion criteria to enter the systematic review.
Reviewer 2 Report
I consider that the objective of this review seems interesting and can contribute additional information to the existing knowledge of this matter. Ventilatory fatigue is one of the major limiting factors in physical performance and therefore the training of the respiratory muscles is a practice of great interest.
It is true that this systematic review has many limitations, but still remains interesting. Perhaps, the discussion is excessively long and complex, which makes it difficult to read and understand it, should try to reduce it as much as possible.
There are some points that should be considered:
Materials and methods
This section should be improved and clarified. The authors should consider providing more in-depth information of the gaps in the present Materials and methods section.
2.3. Search strategy – Data sources
- I have concerns about the completeness of the search strategy. The authors should have consulted more databases to assure a more deep and complete review, such as Medlars International Literature Online (MEDLINE) via PubMed.
- What was the search equation defined with the terms related?
- The authors limited their search to articles from 2000 to 2020, but do not provide a sufficient defense for this decision.
.5. Data collection and extraction
- Readers may not all be familiar with the PEDro scale. Please give more details about the types of bias assessed with this tool. Without having at least one example of a question from each method, it is difficult to assess the meaning of the scales.
- Which authors evaluated the risk of bias of selected articles? Their names or initials should be specified and at least two author should did it.
- Why is not specified the Score breakdown of each evaluated article? The authors only feature an average percentage of all the articles.
- You evaluated the Risk of Bias for each study, but did not discuss how this influenced your interpretation of the results. Do you consider any of the studies (or main outcomes) to be of particularly high risk of bias?
After the inclusion criteria or in a new section, some information should be included regarding how the inclusion criteria were applied? which authors carried them out?
Minor points:
Pag. 16 line 283-284: “All subjects experienced an increase in MEP, ranging from a 9% to 32% improvement, but the increase in maximum expiratory pressure (MEP)”. The meaning of the abbreviation must go alongside the first time the abbreviations appear.
Finally, I consider that the inclusion criteria have given rise to a very large heterogeneity in the chosen studies and that, although it has allowed us to obtain more studies, makes the discussion excessively complex. Perhaps, one should try to be more precise in some inclusion criteria, which could reduce the discussion by eliminating excessive comparisons. For example, considering only professional athletes and/or focusing only on the two major sports, rugby and soccer. This would reduce the number of studies, but it would avoid in the discussion so many comparisons between so many parameters and perhaps it would be less complex to understand them.
Author Response
Dear Reviewer,
Thank you very much for your comments and for the details that you have commented and we have modified.
In addition, at the request of other reviewers we have conducted a review of English. We attach certificate.
In relation to this, we have modified many of the pages of the previous article, so you will find your comment in the main text as reviewer 2.
2.3. Search strategy – Data sources
- I have concerns about the completeness of the search strategy. The authors should have consulted more databases to assure a more deep and complete review, such as Medlars International Literature Online (MEDLINE) via PubMed.
- According to our knowledge, MEDLINE is included in Web of Science, so if we search in PubMed we Will get the same results as in WOS.
What was the search equation defined with the terms related?
- Combinando todos los elementos de búsqueda obtuvimos 15 ecuaciones, nosotros entendemos que esto alarga el estudio, pero si usted lo desea podríamos incluir todas estas ecuaciones.
- Combining all the search elements we obtained 15 equations, we understand that this lengthens the study, but if you wish we could include all these equations.
The authors limited their search to articles from 2000 to 2020, but do not provide a sufficient defense for this decision.
- Discúlpenos, cometimos un error, la búsqueda finalmente no estuvo limitada al año 2000, usamos todos los años disponibles hasta el 20 de marzo de 2020.
- Sorry, we made a mistake, the search was finally not limited to the year 2000, we used all available years until March 20, 2020.
- Data collection and extraction
Readers may not all be familiar with the PEDro scale. Please give more details about the types of bias assessed with this tool. Without having at least one example of a question from each method, it is difficult to assess the meaning of the scales.
- Hemos introducido una breve explicación de los aspectos que se valoran en la escala PEDro.
- We have introduced a brief explanation of the aspects that are assessed on the PEDro scale.
Which authors evaluated the risk of bias of selected articles? Their names or initials should be specified and at least two author should did it.
- Olvidamos poner este apartado, nos disculpamos. Ya está solucionado.
- We forgot to put this section; we apologize. Already solved.
-Why is not specified the Score breakdown of each evaluated article? The authors only feature an average percentage of all the articles.
- Hemos introducido el Score Breakdown mediante una tabla.
- We have introduced the Score breakdown using a table
You evaluated the Risk of Bias for each study, but did not discuss how this influenced your interpretation of the results. Do you consider any of the studies (or main outcomes) to be of particularly high risk of bias?
- Consideramos que no, ya que la calidad metodológica de todos los estudios cumple los requisitos para ser incluidos en la revisión sistemática, lo hemos reflejado en la página X.
- We believe not, since the methodological quality of all the studies meets the requirements to be included in the systematic review, we have reflected this on page X.
After the inclusion criteria or in a new section, some information should be included regarding how the inclusion criteria were applied? which authors carried them out?
- Al igual que con el risk of bias, olvidamos aclarar las iniciales de los authors que llevaron a cabo cada parte.
- As with the risk of bias, we forgot to clarify the initials of the authors who carried out each part.
Minor points:
Pag. 16 line 283-284: “All subjects experienced an increase in MEP, ranging from a 9% to 32% improvement, but the increase in maximum expiratory pressure (MEP)”. The meaning of the abbreviation must go alongside the first time the abbreviations appear.
- Done, typographical error
Finally, I consider that the inclusion criteria have given rise to a very large heterogeneity in the chosen studies and that, although it has allowed us to obtain more studies, makes the discussion excessively complex. Perhaps, one should try to be more precise in some inclusion criteria, which could reduce the discussion by eliminating excessive comparisons. For example, considering only professional athletes and/or focusing only on the two major sports, rugby and soccer. This would reduce the number of studies, but it would avoid in the discussion so many comparisons between so many parameters and perhaps it would be less complex to understand them.
- Consideramos su comentario de gran valor. Tiene usted toda la razón en que podríamos haber incluido solo los dos principales deportes, soccer y rugby, ya que hubiésemos eliminado dos artículos, pero al no haber ninguna revisión previa relacionada en modalidades deportivas intermitentes, nos pareció importante reflejar todos los deportes. Muy posiblemente, sea una limitación del estudio.
- We consider your comment to be of great value. You are absolutely right that we could have included only the two main sports, soccer and rugby, as we would have removed two articles. Still, as there was no previous related review on intermittent sports, we felt it was important to reflect all sports. Possibly, it is a limitation of the study.
Reviewer 3 Report
Inspiratory muscle traning in intermitted sports modalieties:a systematic review by
Juan Lorca, Sergio L. Jiménez, Helios Pareja-Galeano, Alberto Lorenzo may be considered for publication after major corrections:
English language correction required for entire manuscript.
Introduction:
Lines 58-59 sentence unclear; “all of this brings with it..”???
Line 91: there are examples of continuous sports but not intermitted sports modalities.
Methods:
Lines 122-127: there is more intermitted sports (according to Sport-specific nutrition: practical strategies for team sports. Holway FE, Spriet LL. J Sports Sci. 2011;29 Suppl 1:S115-25. doi: 10.1080/02640414.2011.605459. Epub 2011 Aug 11.) it looks like author took less than a half of sports disciplines in their search;
Results:
Tables are very chaotic: what was the rule to put the studies in table’s rows? Date, type of sport, type of test? It’s not obvious.
In table 2: should be Table1 as figure 1 is first and then should be table 1 ; in this table authors use word ”socker” (American word for European “football”) but later in the text there are description of footballers – write both names in the table or change for socker players in text.
Table 3 (according to nomenclature in manuscript) there is starting intensity, but later in table only “progression” to what value? (end value of intensity); in 6th column : “no of weeks” there are some empty spaces: not reported or missed by authors ?? in data concerning Najafi 2019 Spanish language description – translate to English, please.
Table 4 :description of the results; If authors divided the table for 3 separate pages, You should place the studies in different order: it would be much easier to compare the data among the studies, if they be grouped by i.e. specific test; But I would rather prefer to change the graphic to put all data on single page.
Discussion:
Line 237 – not “saw” rather “showed” – in this sentence there are “studies” not i.e. “researchers”;
Lines 240-241: It is difficult to compare professional player to amateurs : professional players due to more hours of obligatory training experience less benefits from additional breathing exercises: they are already in very good form; amateurs who spend most of their time “behind the desk” are less trained and additional training is more beneficial in this group – both categories of the players are not starting from the same level so it’s impossible to compare ones to others.
Line 299, 339 – what means indirectly? They respiratory muscles are trained also by other forms of training? In professional athletes level of respiratory muscles performance may be so high that further improvement may be impossible.
Line 311: should rather be: “as the subjects were semi-professional players”
Line 326: “exciting” values ??
References ; in reference 10th bold the Year of study;
Author Response
Dear Reviewer,
Thank you very much for your comments and for the details that you have commented and we have modified.
In addition, at the request of other reviewers we have conducted a review of English. We attach certificate.
In relation to this, we have modified many of the pages of the previous article, so you will find your comment in the main text as reviewer 3.
Introduction:
Lines 58-59 sentence unclear; “all of this brings with it..”???
- We change the sentence for “all of this causes a reduction.”
Line 91: there are examples of continuous sports but not intermitted sports modalities.
- We introduce the two main sports analyzed, football and rugby
Methods:
Lines 122-127: there is more intermitted sports (according to Sport-specific nutrition: practical strategies for team sports. Holway FE, Spriet LL. J Sports Sci. 2011;29 Suppl 1:S115-25. doi: 10.1080/02640414.2011.605459. Epub 2011 Aug 11.) it looks like author took less than a half of sports disciplines in their search;
- We consider your comment to be of great value. We have done a search with the sports that appear in the mentioned article (very interesting, by the way). We have not obtained any different results; we consider that using "athletes" OR "sprint" OR "high-intensity" in the search shows the vast majority of intermittent sports. For example, the search "inspiratory muscle training" and "athletes" in Web of Science gave us studies of handball or rugby.
Results:
Tables are very chaotic: what was the rule to put the studies in table’s rows? Date, type of sport, type of test? It’s not obvious.
- You are absolutely right, we used alphabetical order, a decision that makes no sense. We have now proposed an order based on the date of publication of the various articles.
In table 2: should be Table1 as figure 1 is first and then should be table 1 ; in this table authors use word ”socker” (American word for European “football”) but later in the text there are description of footballers – write both names in the table or change for socker players in text.
- Thank you very much for this correction. As we have used European English words throughout the text, we have changed soccer for football wherever that word appeared.
Table 3 (according to nomenclature in manuscript) there is starting intensity, but later in table only “progression” to what value? (end value of intensity); in 6th column : “no of weeks” there are some empty spaces: not reported or missed by authors ?? in data concerning Najafi 2019 Spanish language description – translate to English, please.
- The final value is always the same as the final value concerning the percentage of 1RM, but the absolute load is modified
- Two studies are acute, so only one test is made. We have cleared it up in the article.
- We translated Najafi´s part, sorry about the mistake
Table 4 :description of the results; If authors divided the table for 3 separate pages, You should place the studies in different order: it would be much easier to compare the data among the studies, if they be grouped by i.e. specific test; But I would rather prefer to change the graphic to put all data on single page.
- We have changed the order, and we have joined tables 2 and 3 in a single table. Putting all the data together in a single graph has been impossible for us due to the large amount of information that appears on it. Also, we have eliminated a column: Variables analyzed
Discussion:
Line 237 – not “saw” rather “showed” – in this sentence there are “studies” not i.e. “researchers”;
- Done, typographical error
Lines 240-241: It is difficult to compare professional player to amateurs : professional players due to more hours of obligatory training experience less benefits from additional breathing exercises: they are already in very good form; amateurs who spend most of their time “behind the desk” are less trained and additional training is more beneficial in this group – both categories of the players are not starting from the same level so it’s impossible to compare ones to others.
- We totally agree with you, obviously, we can´t compare them, and in theory, professional players should experience minor improvements, but in that paragraph I want to make exact just the opposite, that professionals get better results and one possible explanation is that they have been able to use PowerLung, even though nothing has been proven.
Line 299, 339 – what means indirectly? They respiratory muscles are trained also by other forms of training? In professional athletes level of respiratory muscles performance may be so high that further improvement may be impossible.
- As indirect, we wanted to express the same thing that you have described, that dedicating more hours to daily training will have more developed the respiratory muscles, we have clarified that "due to more hours of physical training".
Line 311: should rather be: “as the subjects were semi-professional players”
- Done, typographical error
Line 326: “exciting” values??
- We change “exciting” for “impressive”
References; in reference 10th bold the Year of study;
- It is a chapter of a book, according to the standards of the magazine itself, it should not be in bold. In instructions for Authors, books, and book chapters: 3. Author 1, A.; Author 2, B. Title of the chapter. In Book Title, 2nd ed.; Editor 1, A., Editor 2, B., Eds.; Publisher: Publisher Location, Country, Year; Volume 3, pp. 154–196.
Reviewer 4 Report
This systematic review provided an interesting summary of evidence on the effects of respiratory and inspiratory muscle training on intermittent sports. Although limitations due to the strict inclusion criteria, the authors provided an informative summary of the current evidence on this area of research.
The methodology for SLR has been properly conducted, using a rigorous process.
However, a revision of the entire manuscript is required to improve the quality of the academic and scientific writing.
Several suggestions have been provided here, however authors can find more in the pdf attached. All the yellow highlights require attention.
Pay attention to a better use of the past tense.
Within the text, consider the use of the for Author and colleagues (year), instead of Author et al. (year).
Specific comments
ABSTRACT
Page 1, line 14. I suggest using a word different from “problem”, like impairment or deterioration or similar.
Page 1, line 14-15. “The objective was to evaluate the results obtained with this type of training in intermittent sports modalities”. Firstly, consider writing (the objective of this review). Secondly, “this type of training” seems inappropriate here, since it would be clearer to directly use “inspiratory muscle training”.
Page 1, line 16-21. Several information has been provided regarding the methodology, but less about results. To stay in the words limit, I suggest to reduce the methodology and provide more results.
INTRODUCTION
Page 2, line 60. Consider revising “ways” with “methods”.
Page 2, line 62-63. “…the work is of low resistance at high speed.” It is not very clear.
Page 2, line 63. “both inspiratory and expiratory muscles can be worked,…”. Consider revising worked with activated or involved
Page 2, line 71. Consider revising “previously” with “previous”.
Page 2, line 79-81. “The first, the acute training, is…”. Please start directly with Acute training. However, the entire sentence should be rephrased since it is not clear. Avoid the use of “that is”.
Page 2, line 84. Please keep consistent the use of muscle.
Page 3, line 96. Please revise as follow. “Therefore, the objective of the present study was to carry out a…”
Page 3, line 97. Please revise as follow “…RMT and IMT on intermittent sports modalities…”
METHODS
Page 3, line 101-103. Please revise as follow “The present study was a systematic review…”. However, please consider revising the two sentences for a more scientific writing style.
Page 3, line 119-120. In systematic literature review, this technique is called snowball technique, which can be applied to already included studies or to previous reviews. Therefore, revise the sentence accordingly. In any case “…to find potentially valid studies for study.” is not appropriate.
Page 3, line 140. “…was reflected…” it is not appropriate. “Reported” it might be better.
Page 4, line 134-138. Please describe the interpretation of the score.
RESULTS
A table providing the methodological quality assessment has to be included with the score of each item and final score.
Page 4, line 154-155. Please revise the sentence. Please rephrase in a similar way “Previous review and meta-analysis were reviewed to detect possible studies eligible for the current systematic literature review.”
Page 4, line 156-157. “The full text of 33 articles was evaluated and 23 were excluded for the following reasons:…”
Page 4, line 157-160. Keep the consistent use of the word “article” here and throughout the manuscript, such as 8 articles; 7 articles; 5 articles; 1 article;
Page 6, line 166. The interpretation of the score has to be described in method section.
Page 6, line 172. Please revise as follow “A total sample of 218 participants was obtained from all the articles…”
Page 6, line 173-176. Make a single and clearer sentence.
Page 6, line 178. “Sportpeople” is not appropriate.
Page 8, line 182. Please revise as follow “The initial effort intensity was very similar…”
Page 8, line 185-187. Please revise as follow. “Three types of intervention were used, such as inspiratory muscle training in a chronic protocol (n=7) [32,37,38,48,58-60], in an acute protocol before the evaluation test (n=2) [46,47]; and with a combination of chronic and acute protocol (n=1) [50].”
Page 8, line 189-191. Please rephrase.
Page 8, line 192-193. Please rephrase and use the past tense.
Page 11, line 201-202. Please revise as follow “The level 1 Yo-Yo test was used as an evaluation test to compare pre- and post-inspiratory muscle training effects in 6 out of 10 articles.”
Page 11, line 206-211. Please revise as follow “A significant increase in MIP was obtained in all studies, ranging from 8% to 33% of improvement. Rate of perceived exertion (RPE) and dyspnea (RPB) significantly decreased in the studies investigated. Nine of the 10 articles showed a significant improvement in performances for their respective assessment tests, referring to a reduction in sprinting time, a greater recovery capacity, and a greater distance covered or speed in sprinting.”
DISCUSSION
Page 15, line 218-231. This paragraph seems no well-structured and provided scattered and random information. I suggest revising this section starting again from the purpose of the review and then provide the main findings in a clearer way. Do not start from the analysis of methodological quality. You can mention after the main findings.
Page 15, line 237. Please revise “saw” with “found”.
Page 15, line 243. It is better to refer to expiratory muscles.
Page 15, line 252. Please rephrase the sentence to make it flows better.
Page 15, line 259. Please clearly mention “…conducted a combined acute and chronic protocol.”
Page 15, line 261-262. Please consider revising. “Although the players were not professional football and rugby athletes, they trained on an average of 2-3 hours a day, 4-5 days a week, therefore they could be considered well-trained athletes.”
Page 16, line 283. Spell out maximum expiratory pressure the first time and then use abbreviation if strongly necessary, otherwise keep using the words.
Page 16, line 291-292. Please rephrase the sentence to make it flows better.
Page 16, line 296. Please revise “justification” with “explanation”.
Page 16, line 308-310. Please rephrase the sentence to make it flows better.
Page 17, line 312-313. Please rephrase the sentence to make it flows better.
Page 17, line 313-314. Please revise as follow “The findings from semi-professional or professional participants have to be separately interpreted.”
Page 17, line 331-333. Structure the sentence in a different way avoiding the excessive repetition of “which”.
Page 17, line 549-350. It is quite reductive to refer to central and peripheral fatigue in this way, since the concept of central and peripheral fatigue is very big, no completely clear and fully investigated. Please revise it.
Page 17, line 361-362. Please revise as follow “In the present review, only previous investigations using the pressure threshold devices was with athletes were considered.”
Page 18, line 389-390 / 404-405. Same sentence. Please revise.
Page 19, line 426-427. Please refer to the maintenance of training adaptations.
Author Response
Dear Reviewer,
Thank you very much for your comments and for the details that you have commented and we have modified.
In addition, at the request of other reviewers we have conducted a review of English. We attach certificate.
In relation to this, we have modified many of the pages of the previous article, so you will find your comment in the main text as reviewer 4.
ABSTRACT
Page 1, line 14. I suggest using a word different from “problem”, like impairment or deterioration or similar.
- We have changed “problem” for “impairment.”
Page 1, line 14-15. “The objective was to evaluate the results obtained with this type of training in intermittent sports modalities”. Firstly, consider writing (the objective of this review). Secondly, “this type of training” seems inappropriate here, since it would be clearer to directly use “inspiratory muscle training”.
- We have changed both corrections.
Page 1, line 16-21. Several information has been provided regarding the methodology, but less about results. To stay in the words limit, I suggest to reduce the methodology and provide more results.
- We have modified the abstract, giving more importance to the results
INTRODUCTION
Page 2, line 60. Consider revising “ways” with “methods”.
- Done
Page 2, line 62-63. “…the work is of low resistance at high speed.” It is not very clear.
- We have provided some clarifications
Page 2, line 63. “both inspiratory and expiratory muscles can be worked,…”. Consider revising worked with activated or involved
- We have changed “worked” for “involved2
Page 2, line 71. Consider revising “previously” with “previous”.
- Done, typographical error
Page 2, line 79-81. “The first, the acute training, is…”. Please start directly with Acute training. However, the entire sentence should be rephrased since it is not clear. Avoid the use of “that is”.
- We rephrased the sentence for Acute training is a specific protocol that is carried out only before starting the tests
Page 2, line 84. Please keep consistent the use of muscle.
- Done, typographical error
Page 3, line 96. Please revise as follow. “Therefore, the objective of the present study was to carry out a…”
- Done
Page 3, line 97. Please revise as follow “…RMT and IMT on intermittent sports modalities…”
- Done
METHODS
Page 3, line 101-103. Please revise as follow “The present study was a systematic review…”. However, please consider revising the two sentences for a more scientific writing style.
- We have provided some clarifications and have changed the sentence
Page 3, line 119-120. In systematic literature review, this technique is called snowball technique, which can be applied to already included studies or to previous reviews. Therefore, revise the sentence accordingly. In any case “…to find potentially valid studies for study.” is not appropriate.
- We have introduced a new sentence talking about snowball sampling
Page 3, line 140. “…was reflected…” it is not appropriate. “Reported” it might be better.
- Done
Page 4, line 134-138. Please describe the interpretation of the score.
- We have introduced an explanation of the PEDro score
RESULTS
A table providing the methodological quality assessment has to be included with the score of each item and final score.
- We have made a table providing the methodological quality assessment
Page 4, line 154-155. Please revise the sentence. Please rephrase in a similar way “Previous review and meta-analysis were reviewed to detect possible studies eligible for the current systematic literature review.”
- Done
Page 4, line 156-157. “The full text of 33 articles was evaluated and 23 were excluded for the following reasons:…”
- We have changed “A total of 33 studies were evaluated in full text, of which 23 were discarded, for the following reasons” for The full text of 33 articles was evaluated and 23 were excluded for the following reasons
Page 4, line 157-160. Keep the consistent use of the word “article” here and throughout the manuscript, such as 8 articles; 7 articles; 5 articles; 1 article;
- Done
Page 6, line 166. The interpretation of the score has to be described in method section.
- Thank you very much for this comment, you were totally right. We have eliminated this section and moved to the method section.
Page 6, line 172. Please revise as follow “A total sample of 218 participants was obtained from all the articles…”
- Done
Page 6, line 173-176. Make a single and clearer sentence.
- We have changed “The studies focused mainly on one sport, football, since eight of the ten articles included in the systematic review have subjects who practice this sport. The second most present was rugby, which is studied in five papers” for “The most studied sports were soccer and rugby, although badminton, field hockey, and basketball were also studied to a lesser extent”
Page 6, line 178. “Sportpeople” is not appropriate.
- We have changed “sportpeople” for “players”
Page 8, line 182. Please revise as follow “The initial effort intensity was very similar…”
- Done
Page 8, line 185-187. Please revise as follow. “Three types of intervention were used, such as inspiratory muscle training in a chronic protocol (n=7) [32,37,38,48,58-60], in an acute protocol before the evaluation test (n=2) [46,47]; and with a combination of chronic and acute protocol (n=1) [50].”
- Done
Page 8, line 189-191. Please rephrase.
- We have changed “On the other hand, the two articles that carried out an acute protocol, carried out the same procedure exactly, with 40% of the MIP, thirty inspirations and two sessions before the test, adding up to a total of sixty inspirations in the warm-up” for “Acute training studies made a 40% of the MIP, thirty inspirations and two sessions before the test”.
Page 8, line 192-193. Please rephrase and use the past tense.
- We have changed “Although the number of inspirations per session is always the same, 30 inspirations except in one article [37], there is much heterogeneity in the number of weekly sessions” for “The number of weekly sessions was very heterogeneous”
Page 11, line 201-202. Please revise as follow “The level 1 Yo-Yo test was used as an evaluation test to compare pre- and post-inspiratory muscle training effects in 6 out of 10 articles.”
- Done
Page 11, line 206-211. Please revise as follow “A significant increase in MIP was obtained in all studies, ranging from 8% to 33% of improvement. Rate of perceived exertion (RPE) and dyspnea (RPB) significantly decreased in the studies investigated. Nine of the 10 articles showed a significant improvement in performances for their respective assessment tests, referring to a reduction in sprinting time, a greater recovery capacity, and a greater distance covered or speed in sprinting.”
- Done
DISCUSSION
Page 15, line 218-231. This paragraph seems no well-structured and provided scattered and random information. I suggest revising this section starting again from the purpose of the review and then provide the main findings in a clearer way. Do not start from the analysis of methodological quality. You can mention after the main findings.
- We have changed the orders of the paragraph and introduced “the objective of the present study was to evaluate the effects of respiratory muscle training (RMT) and inspiratory muscle training (IMT) on intermittent sports modalities”
Page 15, line 237. Please revise “saw” with “found”.
- We have changed “saw” for “showed”
Page 15, line 243. It is better to refer to expiratory muscles.
- Done
Page 15, line 252. Please rephrase the sentence to make it flows better.
- We have modified “Besides, other variables also improved significantly in both cases, such as MIP and shortness of breath” for “Other variables like MIP and dyspnea also improved significantly”
Page 15, line 259. Please clearly mention “…conducted a combined acute and chronic protocol.”
- Done
Page 15, line 261-262. Please consider revising. “Although the players were not professional football and rugby athletes, they trained on an average of 2-3 hours a day, 4-5 days a week, therefore they could be considered well-trained athletes.”
- Done
Page 16, line 283. Spell out maximum expiratory pressure the first time and then use abbreviation if strongly necessary, otherwise keep using the words.
- Done, we have modified this in all the study
Page 16, line 291-292. Please rephrase the sentence to make it flows better.
- We have changed “It would have been relevant that the study by Nicks et al. [59], which used a PowerLung device, which presents a load on both the inspiratory and expiratory muscles, would have provided data of the MEP, to assess the two” for “Only one study used PowerLung, a device that presents a load on both the inspiratory and expiratory phases [59]. However, since it did not provide MEP data, a comparison couldn´t be made”
Page 16, line 296. Please revise “justification” with “explanation”.
- Done
Page 16, line 308-310. Please rephrase the sentence to make it flows better.
- We have changed “It is interesting to note that studies with acute training protocols did not observe a reduction in RPE, while the vast majority of chronic studies that have analyzed it have observed very significant decreases, from 8% [50] to 29% [37]” for “While the majority of chronic studies that have analyzed RPE have observed very significant decreases, from 8% [50] to 29% [37], with a large effect size (ES) (-1.47), acute training protocols did not observe a reduction”
Page 17, line 312-313. Please rephrase the sentence to make it flows better.
- We have deleted this phrase, it did not contribute anything
Page 17, line 313-314. Please revise as follow “The findings from semi-professional or professional participants have to be separately interpreted.”
- Done
Page 17, line 331-333. Structure the sentence in a different way avoiding the excessive repetition of “which”.
- We have changed “The reasons for this improvement in both the Yo-Yo and the RSA tests in semi-professionals and professionals could be due to three main factors. Firstly, a reduction in metaboreflex, which would allow these athletes to get more oxygen to the peripheral muscles, which in turn would lead to a decrease in fatigue and lactate concentration in these muscles. The second reason would be a decrease in the RPE, while the third would be a decrease in the RPB, both of which are related” for “There are three main factors for this improvement in both the Yo-Yo and the RSA tests in semi-professional and professional athletes. Firstly, a reduction in metaboreflex, which would allow these athletes to get more oxygen to the peripheral muscles, secondly a decrease in the RPE, and finally a reduction in the RPB”.
Page 17, line 549-350. It is quite reductive to refer to central and peripheral fatigue in this way, since the concept of central and peripheral fatigue is very big, no completely clear and fully investigated. Please revise it.
- You were right, we have eliminated this phrase
Page 17, line 361-362. Please revise as follow “In the present review, only previous investigations using the pressure threshold devices was with athletes were considered.”
- Done
Page 18, line 389-390 / 404-405. Same sentence. Please revise.
- We have eliminated the sentence of pages 404-405
Page 19, line 426-427. Please refer to the maintenance of training adaptations.
- Done
Round 2
Reviewer 2 Report
I consider that the authors have correctly modified the article.
And, although I consider that the inclusion criteria should have been limited a little more, I also consider that the modifications and explanations given improve and allow a better reading and understanding of the work.
Reviewer 4 Report
The authors satisfied all the requirements.